# Clinical and Genetic Characteristics of the Heidenhain Variant of Creutzfeldt–Jakob Disease

**DOI:** 10.3390/v15051092

**Published:** 2023-04-29

**Authors:** Yu Kong, Zhongyun Chen, Jing Zhang, Xue Wang, Liyong Wu

**Affiliations:** 1Department of Neurology, Xuanwu Hospital, Capital Medical University, Beijing 100053, China; 2Department of Library, Xuanwu Hospital, Capital Medical University, Beijing 100053, China

**Keywords:** Heidenhain variant, visual symptom, Creutzfeldt–Jakob disease, *PRNP*

## Abstract

Background: The Heidenhain variant of Creutzfeldt–Jakob disease (HvCJD), as a rare phenotype of CJD, has been under-recognized. We aim to elucidate the clinical and genetic features of HvCJD and investigate the differences of clinical features between genetic and sporadic HvCJD to improve our understanding of this rare subtype. Method: HvCJD patients admitted to the Xuanwu Hospital from February 2012 to September 2022 were identified, and published reports on genetic HvCJD cases were also reviewed. The clinical and genetic features of HvCJD were summarized, and the clinical features between genetic and sporadic HvCJD were compared. Results: A total of 18 (7.9%) HvCJD patients were identified from 229 CJD cases. Blurred vision was the most common visual disturbance at the disease’s onset, and the median duration of isolated visual symptoms was 30.0 (14.8–40.0) days. DWI hyperintensities could appear in the early stage, which might help with early diagnosis. Combined with previous studies, nine genetic HvCJD cases were identified. The most common mutation was V210I (4/9), and all patients (9/9) had methionine homozygosity (MM) at codon 129. Only 25% of cases had a family history of the disease. Compared to sporadic HvCJD, genetic HvCJD cases were more likely to present with non-blurred vision visual symptoms at onset and develop cortical blindness during the progression of the disease. Conclusions: HvCJD not only could be sporadic, but also, it could be caused by different *PRNP* mutations. Sporadic HvCJD was more likely to present with blurred vision visual symptoms at onset, and genetic HvCJD was more likely to develop cortical blindness with the disease’s progression.

## 1. Introduction

Creutzfeldt–Jakob disease (CJD) is a fatal neurodegenerative disease caused by the deposition of abnormal cellular prion proteins (PrP^Sc^), and it consists of sporadic, genetic, and acquired forms. The clinical presentations of CJD are highly heterogeneous, with rapidly progressive dementia being the typical symptom. The Heidenhain variant, CJD (HvCJD), is a peculiar clinical phenotype characterized by isolated visual disturbances at the disease’s onset without other manifestations [1] and accounts for 3.7–24% of the total number of CJD cases [2,3,4,5]. A few HvCJD cases were reported successively, but most of them were case reports or small case series [4,6,7,8,9,10]. Thus, the overall understanding of HvCJD is still insufficient, and HvCJD is easily misdiagnosed as an ophthalmic disease, resulting in unnecessary surgery and increasing the risk of iatrogenic transmission.

Previous studies have shown that most cases of HvCJD are sporadic, and some have even ascribed it as a special subtype of sporadic CJD [4,11]. However, there are a few case reports of HvCJD patients harboring prion protein gene (*PRNP*) mutations, such as E196A, T188K, V180I, and V210I [12,13,14,15]. A recent study showed that up to 20% of HvCJD cases are genetic [2]. Compared to sporadic HvCJD, the genetic form of HvCJD is associated with a significantly longer survival duration [2], but due to the small sample size of genetic HvCJD, a future study will need to be conducted to verify this finding. Based on current studies, we hypothesized that genetic HvCJD is not uncommon, and mutations in the *PRNP* gene may affect the clinical features of HvCJD. However, there is still a lack of systematic descriptions of genetic HvCJD.

To this end, we reviewed our database of CJD cases collected over the past ten years to summarize the clinical characteristics of HvCJD. Furthermore, we also summarized the clinical features of genetic HvCJD included both in our cohort and previous literature and compared the clinical, auxiliary, and prognostic features of genetic HvCJD with those of sporadic HvCJD cases. The aim is to improve our understanding of this peculiar clinical subtype and guide the disease’s management and genetic counseling.

## 2. Methods

### 2.1. Participants

A total of 229 patients with definitive or probable CJD admitted to Xuanwu Hospital were consecutively recruited from February 2012 to September 2022. Patients enrolled before January 2018 were retrospective cases, and the requirement of informed consent was waived by our ethics committee. Patients admitted to our hospital after February 2018 were prospectively included, and written informed consent was obtained from patients or their next of kin. The study was approved by the ethics committee of Xuanwu Hospital and performed in compliance with ethical principles of the Declaration of Helsinki. 

CJD was diagnosed according to the updated diagnostic criteria for CJD published in 2009 and validated by the current the WHO’s criteria [16,17]. Heidenhain variant was defined as isolated visual symptoms at the disease’s onset from several weeks to several months. Patients diagnosed with possible alternative diagnoses, such as encephalitis, neoplastic-related diseases, and inherited metabolic diseases, and those with improvement in clinical symptoms during the follow-up were excluded. 

PubMed, Embase, and Web of Science databases were searched for research articles and case studies reporting genetic HvCJD patients published on or before December 2022 using the following terms: (“Creutzfeldt-Jakob disease”) AND (“Heidenhain”). Studies reporting HvCJD cases that had *PRNP* mutations were selected. Publications for which patients’ original clinical data were not reported and case series in which patients’ characteristics were not available separately were excluded. A total of 397 articles were retrieved in the initial search. After the removal of duplicates by analyzing titles and abstracts, the full texts of 23 articles were screened, and 5 articles reporting six patients were finally included in this study [12,13,14,15,18].

### 2.2. Clinical Data Collection

Epidemiological and clinical information, including sex, age at onset, familial history, symptoms and signs, auxiliary examinations and survival time, were extracted for all patients. The initial visual symptoms and those manifesting during the disease’s course, including blurred vision, diplopia, visual hallucination, cortical blindness, hemianopsia, visuospatial dysfunction, metamorphopsia and dyschromatopsia, were analyzed separately. Other non-visual neurological manifestations included cognitive decline, psychiatric symptoms, myoclonus, pyramidal signs, extrapyramidal signs, cerebellar signs, sensory disturbance, a speech disorder and akinetic mutism. Auxiliary tests included optical coherence tomography (OCT), visual evoked potential (VEP), visual field test, cerebrospinal fluid (CSF) 14-3-3 protein, brain magnetic resonance imaging (MRI), electroencephalogram (EEG), real-time quaking-induced conversion assay (RT-QuIC), *PRNP* mutations and codon 129 genotypes. The survival duration was defined as the time from the first symptom onset to death.

### 2.3. Laboratory Tests

Genomic DNA was extracted from fresh peripheral blood leukocytes, and whole exome sequencing (WES) libraries were generated using the Agilent SureSelect Human All Exon V6 Kit (Agilent Technologies, Santa Clara, CA, United States), and then sequenced as described previously [19]. CSF samples were collected at our hospital and sent to the Chinese Center for Disease Control and Prevention (CDC) to detect CSF 14-3-3 protein via Western blotting [20]. EEG was performed using a 21-channel digital EEG system (Micromed, Italy). PSWC was defined according to the criteria published in 1996 [21]. Brain MRI scans were performed on a 3.0T MRI scanner (Erlangen, Germany) and T1-weighted image (T1WI), T2-weighted image (T2WI), fluid-attenuated inversion recovery (FLAIR), diffusion-weighted imaging (DWI), and diffusion coefficient (ADC) results were acquired. The typical MRI features were high signals in caudate/putamen or at least two cortical regions (temporal, parietal and occipital) either via DWI or FLAIR. The CSF RT-QuIC assay was performed at the Chinese CDC using the protocol as described previously [22]. PET scans were performed on the GE Signa PET/MR 3.0 Tesla scanner (GE Healthcare Milwaukee, WI) using 18F-FDG (~308 MBq) to assess the metabolism of brain.

### 2.4. Statistical Analysis

Continuous variables are presented as mean ± standard deviation (SD) or median and interquartile range (IQR) depending on whether they conformed to normal distribution. Categorical variables are presented as the frequency (percentage). The Mann–Whitney U test or *t*-test was used to compare continuous variables. The Chi-square test or Fisher’s exact test was used to compare categorical variables. A two-tailed *p*-value ≤ 0.05 was considered to be statistically significant. All statistical analyses were performed using SPSS, version 25.

## 3. Results 

### 3.1. The Clinical Features of HvCJD Cases Included in Our Cohort

A total of 229 patients with definite and probable CJD were admitted to our hospital, of which 18 (7.9%) cases presented with the Heidenhain variant phenotype. Detailed clinical data are summarized in Table 1. Eight HvCJD patients were female (44.4%), and ten (55.6%) were male. The mean age at the disease’s onset was 61.8 (7.7) years. Three HvCJD patients harbored a *PRNP* mutation (1 T188K, 1 E200K and 1 V203I), suggesting that genetic HvCJD accounted for 16.7% of the total HvCJD cases. All patients had the MM homozygote at codon 129. At the time of data analysis, 13 patients had died, and the median survival time was 7.0 (4.0–16.0) months.

Blurred vision was the most common initial visual disorder, which accounted for 83.3% of HvCJD patients. In addition, two patients started with diplopia, and one patient started with metamorphopsia. The median duration of isolated visual symptoms was 30.0 (14.8–40.0) days, ranging from 7 to 120 days. During the entire disease’s course, blurred vision (88.9%) was also the most common symptom, followed by visual hallucinations (27.8%), diplopia (16.7%), metamorphopsia (16.7%), hemianopsia (11.1%), visuospatial dysfunction (5.6%), dyschromatopsia (5.6%) and cortical blindness (5.6%). Among the first non-visual neuropsychological symptoms that followed initial visual impairment, cognitive decline (44.4%) was the most common one, followed by unsteady gait (33.3%), psychiatric symptoms (5.6%), a speech disorder (5.6%), sensory disturbances (5.6%) and extrapyramidal signs (5.6%).

All HvCJD cases underwent brain MRI scans, and 14 (77.8%) cases were performed more than twice. Only 55.6% of patients fulfilled the WHO’s criteria for CJD in the initial scans, and this proportion was up to 88.9% in the serial scans. The median time from the disease’s onset to the presence of typical neuroimaging manifestations of CJD was 2.0 (1.0–3.0) months. However, not all HvCJD (77.8%) patients had restricted diffusion in the occipital lobe. Even though some patients underwent more than one EEG, PSWCs were identified in only 61.1% of patients. The median time from the disease’s onset to the first observation of PSWCs was 2.5 (2.0–3.0) months. In addition, CSF 14-3-3 protein analysis was performed in fourteen patients, and nine (64.3%) had a positive result. The median time from the disease’s onset to positive CSF 14-3-3 protein was 2.5 (1.8–4.0) months. The CSF RT-QuIC assay was only performed on one patient, and the result showed positive PrP^Sc^ seeding activity. 

### 3.2. The Clinical Features of Genetic HvCJD Cases

Apart from three genetic HvCJD enrolled from our center, we also identified six genetic HvCJD cases from previously published reports. The detailed clinical features of a total of nine genetic HvCJD are shown in Table 2. Five patients (55.6%) were female, and four (44.4%) were male. The mean age at the disease’s onset was 61.6 (10.2) years, and the median survival duration was 12.5 months (2–19 months). Only two patients (25%) had a family history of HvCJD, and both (case 5 and case 6) were from the same family. Among all nine genetic HvCJD cases, four patients from three families carried the V210I mutation, and the other five carried T188K, E200K, V203I, V180I and E196A, respectively. All of these mutations are located in the C-terminal domain of *PRNP* (Figure 1). In addition, all of these patients had methionine homozygosity (MM) at codon 129. 

The most common initial visual symptom was metamorphopsia (3/9, 33.3%), followed by blurred vision (2/9, 22.2%), diplopia (2/9, 22.2%), visuospatial dysfunction (1/9, 11.1%), hemianopsia (1/9, 11.1%), dyschromatopsia (1/9, 11.1%) and visual hallucinations (1/9, 11.1%). In addition, 75% of the patients developed non-visual symptoms within one month of the onset. The longest duration of isolated visual symptoms was 120 days. The most common visual symptom during the entire disease course was also metamorphopsia (4/9, 44.4%), followed by blurred vision (3/9, 33.3%), visual hallucinations (3/9, 33.3%), cortical blindness (3/9, 33.3%), diplopia (2/9, 22.2%), visuospatial dysfunction (2/9, 22.2%), hemianopsia (1/9, 11.1%) and dyschromatopsia (1/9, 11.1%).

The median time from the disease’s onset to the first brain MRI was 1 month (0.5–7), and 55.6% (5/9) of the patients presented the typical DWI hyperintensity of CJD. Except for one patient without DWI hyperintensity who underwent brain MRI only once, all patients exhibited this feature on serial MRI scans. The brain MRI manifestations of three patients enrolled in our hospital are shown in Figure 2. The median time to the first EEG was 1.25 months (0.7–7), and 37.5% (3/8) of the patients presented PSWCs. The positive rate of PSWCs was up to 50% in the serial EEG examinations. Seven patients were analyzed for 14-3-3 protein in the CSF, and six (85.7%) had positive results. CSF RT-QuIC was performed on four patients, and three (75%) showed PrP^Sc^ seeding activity. Only case 1 underwent the FDG PET scan and showed severe hypermetabolism in the bilateral parietal and occipital lobes. Although visual disturbances were the initial symptoms, only two patients underwent visual-related examinations. One patient had normal OCT and prolonged latencies as per VEP, and the other had an abnormal visual field test result.

### 3.3. The Differences between Sporadic and Genetic HvCJD

The clinical, accessory, and prognostic differences between the 9 genetic HvCJD patients, and 15 sporadic HvCJD patients (all from our in-home database) were compared. As shown in Table 3, sex and age at onset were similar between the genetic and sporadic HvCJD groups (*p* = 0.675 and *p* = 0.950, respectively). Blurred vision was more common as the initial visual symptom in sporadic HvCJD (86.7% and 22.2%, respectively, *p* = 0.003), while genetic HvCJD patients were more likely to present with other visual disorders, such as diplopia, visuospatial dysfunction, hemianopsia, visual hallucinations, metamorphopsia and dyschromatopsia. The median duration of isolated visual symptoms was similar between the two groups (*p* = 0.925). Throughout the disease’s course, blurred vision was also more frequent among sporadic HvCJD patients compared to its frequency among the genetic HvCJD patients (93.3% and 33.3%, respectively, *p* = 0.004), while the latter group were more likely to develop cortical blindness than patients with sporadic HvCJD were (33.3% and 0%, respectively, *p* = 0.042). Pyramidal signs were more common among sporadic HvCJD patients than it was among those with genetic HvCJD (73.3% and 22.2%, respectively, *p* = 0.033). All patients enrolled in our hospital were followed up every three months, and at the time of data analysis, 6 genetic HvCJD patients and 13 sporadic HvCJD patients had died. Although the survival duration of genetic HvCJD cases was longer than that of sporadic HvCJD cases, the result did not reach statistical significance (*p* = 0.525).

## 4. Discussion

In this study, we described the clinical and genetic characteristics of HvCJD and investigated the clinical similarities and differences between genetic subtypes and sporadic subtypes. Among the CJD patients included in our center, 7.9% of the cases presented with the Heidenhain variant phenotype. Blurred vision was the most common symptom at the disease’s onset, and non-visual neuropsychiatric symptoms could appeare after an average of one month. DWI hyperintensity sometimes occurred at the early disease stage. A total of 16.7% of HvCJD cases were genetic, with the most common mutation being V210I, but only 25% of cases had a family history of the disease. Compared to sporadic HvCJD, genetic HvCJD patients were more likely to present with non-blurred vision at the disease’s onset and develop cortical blindness during the disease’s course. Our study describes the clinical and genetic features of HvCJD in a systematic manner, which can aid in clinical diagnosis and genetic counseling. 

Visual disturbances are common symptoms during the course of CJD, but they are relatively rare isolated symptoms at the disease’s onset. In our study, HvCJD accounted for 7.9% of all CJD cases, which is consistent with previous studies [4,5]. Blurred vision was the most common visual symptom at the disease’s onset. With the increase in age, people may develop blurred vision due to the aging of ocular structures, especially among patients in their 50s, which is also the common onset age of CJD. Thus, this symptom may be overlooked, and a proportion of patients may visit the ophthalmology clinic. A previous study revealed that 35% of HvCJD cases presented to ophthalmologists were misdiagnosed with vitreous opacity, macular degeneration, or cataracts [2]. This can delay treatment and increase the risk of iatrogenic transmission via surgery. Thus, for patients with visual impairment, families and clinicians should pay attention to observing other non-visual symptoms of patients. Our study combined previous studies and showed that non-visual symptoms could occur one week after the appearance of visual symptoms [2] and cognitive decline and unsteady gait are the most common ones. In the case of other neurologic symptoms, the patient should visit a neurology department for a further evaluation.

Ancillary tests can help clinicians differentiate CJD from other neurological diseases and ophthalmic disorders. Brian MRI is an essential tool in the diagnosis of CJD, and our study showed that the sensitivity of MRI was superior to those of CSF 14-3-3 and EEG, which is consistent with previous studies [23,24,25,26]. In our study, the median time from the disease’s onset to the presence of typical neuroimaging manifestations of CJD was 2 months. Some data showed that DWI hyperintensity could occur in a very early stage and even before the symptoms’ onset [27,28,29]. Therefore, we recommend that patients with visual impairment that cannot be attributed to ocular diseases or those whose etiology is unclear undergo brain MRI to rule out intracranial disease. For patients with a suspected diagnosis of CJD, we recommend further performing a CSF RT-QuIC assay. CSF RT-QuIC is a disease-specific biomarker and has high diagnostic accuracy, with the specificity being 99–100% [30]. This technique has greatly improved the accuracy of the pre-mortem diagnosis of CJD and is of great significance for the early diagnosis of patients with atypical clinical phenotypes.

HvCJD is usually regarded as a peculiar phenotype of sporadic CJD, and only a few cases of genetic HvCJD have been reported so far. We found that genetic HvCJD accounted for 16.7% of all HvCJD cases, and Yang et al. [2] similarly reported *PRNP* mutations in 20% of the Heidenhain variants, indicating that genetic HvCJD is not very rare. Interestingly, only two (25%) cases in our cohort had a family history of the disease, which could be attributed to the occurrence of *de novo* mutations, as well as the low penetrance of genetic CJD. For instance, the penetrance rates of V210I and V180I are ~10% and 1%, respectively [31]. Thus, *PRNP* gene tests should be routinely performed in HvCJD cases, even in the absence of family history. 

At present, six *PRNP* gene mutations have been detected in HvCJD so far, including V180I, T188K, E196A, E200K, V203I, and V210I. All these mutations occur in the 𝛼2–𝛼3 helixes of the C-terminal domain sequence of the *PRNP* gene [32]. The underlying mechanism is unclear, and we hypothesize that it may be related to the fact that most genetic CJD-related *PRNP* mutations are located in the C-terminal region [33]. In addition, most HvCJD cases are methionine homozygous (MM) at codon 129 of PrP^Sc^. In one study conducted on 16 patients with sporadic HvCJD, all cases had MM at codon 129^4^, which is consistent with our findings. Thus, the Heidenhain variant phenotype might be related to specific changes in the *PRNP* gene, although the mechanisms need further investigation.

V210I was the most common mutation among patients with genetic HvCJD, and it was present in 44.4% of the cases. It is also the most common *PRNP* mutation in Italy [34]. The clinical manifestations of genetic CJD with V210I mutation are similar to those of sCJD [33,35]. In a German cohort, 15% of the patients with V210I mutation presented visual symptoms at the disease’s onset, and 85% of them developed visual symptoms during the disease’s course, indicating the susceptibility of the visual cortex to the V210I mutation [33,35,36]. We also detected E200K and V203I mutations in the Heidenhain variant phenotype, thus expanding the genetic spectrum of HvCJD. E200K is the most common *PRNP* mutation that causes genetic CJD worldwide [31,35]. The most common initial symptom associated with this mutation is cognitive decline (50%), followed by cerebellar ataxia (30%) and sensory (15%) and vegetative symptoms (5%) [36]. Our study shows that genetic CJD with E200K can also manifest as a visual disturbance at the onset. In addition, due to the low penetrance and lack of segregation in families, V203I has been considered as a risk genotype for CJD without Mendelian inheritance [31,37]. Analysis of the clinical data of 10 CJD patients with V203I variant revealed that the most common symptom was cognitive decline [37]. Thus, CJD should be considered for patients carrying the V203I mutation and exhibiting visual disorders, and relevant auxiliary tests should be conducted to rule out the disease. 

There were some differences between the clinical features of genetic HvCJD and sporadic HvCJD in our cohort. While blurred vision was more common in sporadic HvCJD patients as the initial visual symptom, other visual symptoms were more frequent in patients with genetic HvCJD. Thus, HvCJD patients with non-blurred vision should be screened for *PRNP* gene mutations. In addition, we found that genetic HvCJD patients were more likely to develop cortical blindness, which can trigger psychological disorders including fear, depression, and anxiety, and increase the burden on the family. Therefore, HvCJD patients, especially those with *PRNP* mutations, should be informed of the possibility of cortical blindness and its impact on their quality of life and family, and any changes in their vision should be investigated promptly. However, due to the relatively low number of patients, we could not draw definitive conclusions, and future studies are needed to verify these findings. 

Previous studies have shown that *PRNP* gene mutations are associated with the prolonged survival of patients with HvCJD^2^. Although the survival duration of patients with genetic HvCJD is longer than that of patients with sporadic HvCJD, the difference is not statistically significant. Since the codon 129 was MM in all HvCJD cases included in our study, the effect of polymorphism at codon 129 on survival was excluded. We hypothesized that the differences between our study and the previous study might be related to the different mutations of *PRNP*. Two genetic HvCJD patients in the previous study carried T188K and E196A mutations, while V210I was the most common mutation in our study. Previous studies showed that the disease duration of V210I genetic CJD is reportedly shorter than that of other forms of genetic CJD [35,36]. Three of four patients with the V210I mutation in our cohort were dead within 10 months, which shortened the median survival duration of the genetic HvCJD group. Thus, the survival of HvCJD patients is associated with the specific mutations in *PRNP* gene. 

There are several limitations in our study that ought to be considered. First, the sample size of the genetic HvCJD cases was small, and all enrolled patients were from a single center. Therefore, to assess the clinical features of genetic HvCJD more accurately, we included the cases reported in previous studies. Second, the autopsy rate is very low in China due to traditional ethical values. Thus, all sporadic HvCJD cases in our study were clinically diagnosed without pathological confirmation. Our findings will have to be verified by further studies with large sample sizes. 

## 5. Conclusions

HvCJD not only could be sporadic, but it also could be caused by different *PRNP* mutations. Blurred vision is the most common visual symptom both at the disease’s onset and during the disease course, and brain MRI might help with early diagnosis. In genetic HvCJD, V210I is the most common mutation, and most patients lack a family history of the disease. Compared to sporadic HvCJD, genetic HvCJD is associated with non-blurred vision at the disease’s onset and cortical blindness in the subsequent course of the disease. 

## Figures and Tables

**Figure 1 viruses-15-01092-f001:**
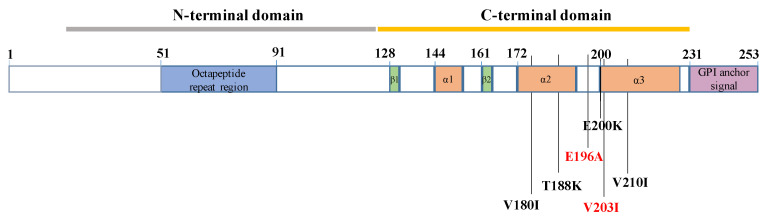
Schematic of PRNP mutations associated with Heidenhain variant phenotypes.

**Figure 2 viruses-15-01092-f002:**
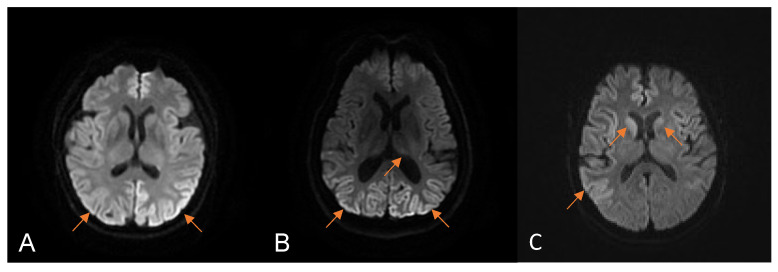
The manifestations of brain magnetic resonance imaging (MRI) in patients with HvCJD enrolled at our hospital. Diffusion-weighted imaging (DWI) MRI of case 1 (**A**) one month after onset showed diffuse cortical hyperintensity. DWI MRI of case 2 (**B**) 7 months after onset showed bilateral occipital cortical and thalamic hyperintensity. DWI MRI of case 3 (**C**) 2 months after onset showed bilateral cortex and head, indicating caudate nucleus hyperintensity. The arrows represent restricted diffusion.

**Table 1 viruses-15-01092-t001:** The clinical data of HvCJD cases included in our center.

Variables	Total HvCJD (N = 18)
Female, n (%)	8 (44.4%)
Age at onset, mean ± SD, years	61.8 ± 7.7
Family history, n (%)	0 (0%)
Duration of isolated visual symptoms, median (IQR), days	30.0 (14.8–40.0)
Survival time, median (IQR), months	7.0 (4.0–16.0)
Initial visual symptom	
Blurred vision, n (%)	15 (83.3%)
Diplopia, n (%)	2 (11.1%)
Metamorphopsia, n (%)	1 (5.6%)
Visual symptoms during the entire course	
Blurred vision, n (%)	16 (88.9%)
Diplopia, n (%)	3 (16.7%)
Visual hallucination, n (%)	5 (27.8%)
Cortical blindness, n (%)	1 (5.6%)
Visuospatial dysfunction, n (%)	1 (5.6%)
Hemianopsia, n (%)	2 (11.1%)
Metamorphopsia, n (%)	3 (16.7%)
Dyschromatopsia, n (%)	1 (5.6%)
The first non-visual symptom	
Cognitive decline, n (%)	8 (44.4%)
Unsteady gait, n (%)	6 (33.3%)
Psychiatric symptoms, n (%)	1 (5.6%)
Sensory disturbance, n (%)	1 (5.6%)
Speech disorder, n (%)	1 (5.6%)
Extrapyramidal signs, n (%)	1 (5.6%)
Auxiliary examinations	
Restricted diffusion in the first MRI, n (%)	13 (72.2%)
MRI fulfilled the WHO criteria for CJD, n (%)	10 (55.6%)
Restricted diffusion on occipital lobe, n (%)	10 (55.6%)
Restricted diffusion in the serial MRI, n (%)	17 (94.4%)
MRI fulfilled the WHO criteria for CJD, n (%)	16 (88.9%)
Restricted diffusion on occipital lobe, n (%)	14 (77.8%)
PSWCs on initial EEG, n (%)	7 (38.9%)
PSWCs on the serial EEG, n (%)	11 (61.1%)
Positive CSF 14-3-3 protein, n/total (%)	9/14 (64.3%)
CSF RT-QuIC, n/total (%)	1/1 (100%)
Hypometabolism on PET, n/total (%)	3/3 (100%)
*PRNP* mutation, n (%)	3 (16.7%)
MM homozygote at codon 129, n (%)	18 (100%)

**Table 2 viruses-15-01092-t002:** The epidemiological, clinical, auxiliary and prognosed data of patients with genetic HvCJD.

Variables	Case 1	Case 2	Case 3	Case 4 [15]	Case 5 [14]	Case 6 [14]	Case 7 [18]	Case 8 [12]	Case 9 [13]
Sex	Female	Male	Female	Female	Male	Female	Male	Female	Male
Age at onset, years	66	57	63	54	59	66	78	69	42
Family history	−	−	−	−	+	+	−	ND	−
Duration of isolated visual symptoms, days	14	120	40	21	21	14	30	28	ND
Survival time, months	4 (alive)	16	18	19	3	9	ND	2	ND
Initial visual symptom									
Blurred vision	+	+	−	−	−	−	−	−	−
Diplopia	−	−	+	+	−	−	−	−	−
Visuospatial dysfunction	−	−	−	−	−	−	−	−	+
Hemianopsia	−	−	−	−	+	−	−	−	−
Metamorphopsia	−	−	−	+	−	+	−	+	−
Dyschromatopsia	−	−	−	−	−	−	−	−	+
Visual hallucination	−	−	−	−	−	−	+	−	−
Visual symptoms during the entire course									
Blurred vision	+	+	−	−	−	−	−	+	−
Diplopia	−	−	+	+	−	−	−	−	−
Visual hallucination	−	−	−	−	+	+	+	−	−
Cortical blindness	+	−	−	−	+	−	−	−	+
Visuospatial dysfunction	−	−	−	−	−	−	+	−	+
Hemianopsia	−	−	−	−	+	−	−	−	−
Metamorphopsia	−	−	−	+	−	+	−	+	+
Dyschromatopsia	−	−	−	−	−	−	−	−	+
Auxiliary examinations									
OCT	−	ND	ND	ND	ND	ND	ND	ND	ND
Visual field test	ND	ND	ND	ND	ND	+	ND	ND	ND
VEP	+	ND	ND	ND	ND	ND	ND	ND	ND
Restricted diffusion in the first MRI	+	+	−	+	−	+	+	−	+
MRI fulfilled the WHO criteria for CJD	−	+	−	+	−	+	+	−	+
Restricted diffusion on occipital lobe	+	+	−	+	−	−	+	−	+
Restricted diffusion on basal ganglia	−	−	−	+	−	+	−	−	−
Restricted diffusion on thalamus	−	+	−	−	−	−	−	−	−
Restricted diffusion in the subsequent MRI	+	+	+	+	ND	ND	ND	+	ND
MRI fulfilled the WHO criteria for CJD	+	+	+	+				+	
Restricted diffusion on occipital lobe	+	+	+	+				−	
Restricted diffusion on basal ganglia	−	−	+	+				+	
Restricted diffusion on thalamus	−	+	−	−				−	
PSWCs on initial EEG	−	−	+	−	+	ND	−	−	+
PSWCs on subsequent EEG	ND	ND	ND	+	ND	ND	ND	ND	ND
Positive CSF 14-3-3 protein	+	+	ND	ND	−	+	+	+	+
RT-QuIC	+	ND	ND	ND	+	+	−	ND	ND
Hypometabolism on PET	+	ND	ND	ND	ND	ND	ND	ND	ND
*PRNP* mutation	T188K	E200K	V203I	V210I	V210I	V210I	V180I	V210I	E196A
MM homozygote at codon 129	MM	MM	MM	MM	MM	MM	MM	MM	MM

+: positive result; −: negative result; ND: not done.

**Table 3 viruses-15-01092-t003:** Comparison of demographic and clinical characteristics between genetic and sporadic HvCJD.

Variables	Genetic HvCJD (n = 9)	Sporadic HvCJD (n = 15)	*p* Value
Female, n (%)	5 (55.6%)	6 (40%)	0.675
Age, mean ± SD, years	61.6 ± 10.2	61.8 ± 8.4	0.950
Family history	2 (22.2%)	0 (0%)	0.111
Initial visual symptom			
Blurred vision, n (%)	2 (22.2%)	13 (86.7%)	0.003
Diplopia, n (%)	2 (22.2%)	1 (6.7%)	0.533
Visuospatial dysfunction, n (%)	1 (11.1%)	0 (0%)	0.375
Hemianopsia, n (%)	1 (11.1%)	0 (0%)	0.375
Metamorphopsia, n (%)	3 (33.3%)	1 (6.7%)	0.130
Dyschromatopsia, n (%)	1 (11.1%)	0 (0%)	0.375
Visual hallucination, n (%)	1 (11.1%)	0 (0%)	0.375
Visual symptoms during the CJD course			
Blurred vision, n (%)	3 (33.3%)	14 (93.3%)	0.004
Diplopia, n (%)	2 (22.2%)	2 (13.3%)	0.615
Visual hallucination, n (%)	3 (33.3%)	5 (33.3%)	1.000
Cortical blindness, n (%)	3 (33.3%)	0 (0%)	0.042
Visuospatial dysfunction, n (%)	2 (22.2%)	1 (6.7%)	0.533
Hemianopsia, n (%)	1 (11.1%)	2 (13.3%)	1.000
Metamorphopsia, n (%)	4 (44.4%)	3 (20%)	0.356
Dyschromatopsia, n (%)	1 (11.1%)	1 (6.7%)	1.000
Non-visual neuropsychiatric symptoms			
Cognitive decline, n (%)	7 (77.8%)	15 (100%)	0.130
Myoclonus, n (%)	6 (66.7%)	10 (66.7%)	1.000
Pyramidal signs, n (%)	2 (22.2%)	11 (73.3%)	0.033
Extrapyramidal signs, n (%)	5 (55.6%)	10 (66.7%)	0.678
Cerebellar signs, n (%)	5 (55.6%)	9 (60%)	1.000
Epilepsy, n (%)	2 (22.2%)	2 (13.3%)	0.615
Psychiatric symptoms, n (%)	5 (55.6%)	4 (26.7%)	0.212
Akinetic mutism, n (%)	5 (55.6%)	5 (33.3%)	0.403
Auxiliary examinations			
Typical hyperintensity on DWI, n (%)	8 (88.9%)	14 (93.3%)	1.000
DWI hyperintensity on occipital lobe, n (%)	6 (66.7%)	11 (73.3%)	1.000
DWI hyperintensity on basal ganglia, n (%)	5 (55.6%)	6 (40%)	0.675
PSWC on EEG, n/total (%)	4/8 (50%)	10/15 (66.7%)	0.657
Positive CSF 14-3-3 protein, n/total (%)	6/7 (85.7%)	7/12 (58.3%)	0.333
MM homozygote at codon 129, n/total (%)	8/8 (100%)	15/15 (100%)	1.000
Duration of isolated visual symptoms, median(IQR), days	24.5 (15.8–37.5)	30.0 (15.0–30.0)	0.925
Survival time, median (IQR), months	12.5 (2.8–18.3)	7.0 (4.0–10.0)	0.525

## Data Availability

The dataset generated and analyzed in the current study is available from the corresponding author on reasonable request.

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
