# Peer review of "Clinical and Genetic Characteristics of the Heidenhain Variant of Creutzfeldt–Jakob Disease"

_viruses, 2023, doi:10.3390/v15051092_

Round 1

Reviewer 1 Report

This is a solid paper on a relatively rare form of sporadic and genetic form of Creutzfeldt-Jakob disease. 

I have 2 remarks:

1. sentence in introduction page 1 line 42-43: 'A recent study... This sentence is not entirely clear. Is here intended to say that 20% of the HvCJD is genetic in origin or is intended to say that of all genetic CJD, 20% has a Heidenhain presentation? I assume the former but it should be clarified/rephrased.

2. Table 1 is quite exhaustive. While it should not be removed in my opinion as it carries specific information such as genotype per case, I believe that the non visual neuropsychiatric symptoms can be removed (it is assumed that these cases all meet the WHO/Eurocjd criteria). Further I believe that of the MRI DWI findings only the presence or absence of occipital changes is interesting, with adding a row if the MRI fulfilled the WHO/Eurocjd criteria for CJD (do not mention all the cerebral fields except the occipital cortex and the final MRI evaluation). Please note that hyperintensity on DWI should be indicated as restricted diffusion.

I believe that these minor changes will improve your paper and its readability. 

Reviewer 2 Report

Thank you for the opportunity to review this interesting article. The aim of the study is to "elucidate the clinical features of genetic HvCJD " and to compare the clinical characteristics of genetic and sporadic HvCJD. The study enrolled 13 sporadic and 3 genetic HvCJD patients seen in the Xuanwu medical center and additional 6 patients were collected from previously published literature. This is the main criticism on this study. The goal of the study is to describe the clinical features of genetic HvCJD but actually provides new data on only 3 patients. It is not clear to me who are the 6 patients that were included in the study since there is no reference to the final 5 articles reporting 6 patients that were included in the study. In the introduction section, there is a paragraph that mutations found in HvCJD with reference to 4 articles. (6-9) Were those patients included in the study?  If this is the case, at least, some of the patients do not meet the inclusion criteria. Heidenhain variant was defined by the authors as "isolated visual symptoms at disease onset for several weeks to several months. The patient reported by Wu et al (Prion 2020) was presented with "progressive dementia, metamorphopsia and change in visual space and visual colour for 2 months" which mean that this patient could not be diagnosed with  Heidenhain variant per the author's definition since he was presented with mixed symptoms (dementia and visual symptoms) no data was provided at this report about the survival time of this patient.  Considering the limited new data on genetic HvCJD in this study, I recommend to change the focus of the study to describe a large series of 18 HvCJD patients and not to focus on genetic HvCJD.  

Minor corrections:

1.       Page 1: abstract -methods: "sporadic and genetic phenotypes"- I recommend to change to subtypes

2.       Page 1- introduction- " and account for only 3.7-24%"- I recommend to omit "only". 24% is a significant proportion of patients.

3.       Page 1- introduction- " and some have even ascribed it ..."- reference.

4.       Page 1- introduction- "the genetic form … is associated with significant longer survival duration"- this conclusion is based on only 2 patients…

5.       Page 8- line 208-2011- he authors suggest that CJD with a mutation in the C terminus tends to involve the posterior region. No explanation was provided why it will accumulate in the posterior region more than other brain regions. Considering the facts that most of the mutations are located at the C terminus (as stated by the authors) and mutations in the C terminus are common in other types of CJD, this speculation is even more problematic.

6.       Page 8- line 227- " our study is the first to show…"-  Zerr et al, Eur J Epidemiol 2016 Feb;31:187-96 already described visual/oculomotor deficits early in the disease of E200K CJD patients  

Round 2

Reviewer 2 Report

Dear editor,

The manuscript had improved with the corrections of most of the minor comments that were addressed. However, the significant problem of a study on genetic HvCJD based on 6 already published  cases and additional new 3 cases. My recommendation was to change the focus of the manuscript to describe the clinical features of  HvCJD- not necessary genetic. Another minor comment that was not addressed is to add in the "methods" section references to the published cases reports that the study is based on.

Author Response

Response: Thanks for your constructive comments. We agree with your comment. The new data on genetic HvCJD is limited and changing the focus to describe a large series of total HvCJD patients makes the study more comprehensive and reasonable. We have revised the full text comprehensively according to your comment. In addition, we have added the references in the “methods” section (…and 5 articles reporting 6 patients were finally included in this study12-15,18.).

Round 3

Reviewer 2 Report

I have no  further comments